# Higher SARS-CoV-2 seroprevalence in workers with lower socioeconomic status in Cape Town, South Africa

**Jane Alexandra Shaw**[1]*, **Maynard Meiring**[1], **Tracy Cummins**[1], **Novel N. Chegou**[1], **Conita Claassen**[1], **Nelita Du Plessis**[1], **Marika Flinn**[1], **Andriette Hiemstra**[1], **Léanie Kleynhans**[1], **Vinzeigh Leukes**[1], **Andre G. Loxton**[1], **Candice MacDonald**[1], **Nosipho Mtala**[1], **Helmuth Reuter**[2], **Donald Simon**[1], **Kim Stanley**[1], **Gerard Tromp**[1], **Wolfgang Preiser**[3], **Stephanus T. Malherbe**[1], **Gerhard Walzl**[1]

**1** DSI-NRF Centre of Excellence for Biomedical Tuberculosis Research, South African Medical Research Council Centre for Tuberculosis Research, Department of Molecular Biology and Human Genetics, Faculty of Medicine and Health Sciences, Stellenbosch University, Cape Town, South Africa, **2** Division of Clinical Pharmacology, Department of Medicine, Stellenbosch University and Tygerberg Academic Hospital, Cape Town, South Africa, **3** Division of Medical Virology, Department of Pathology, Faculty of Medicine and Health Sciences, Stellenbosch University and National Health Laboratory Service, Tygerberg Academic Hospital, Cape Town, South Africa

☯ These authors contributed equally to this work.

* janeshaw@sun.ac.za

**Data Availability Statement:** All relevant data are within the manuscript.

## Abstract

### Background

Inequality is rife throughout South Africa. The first wave of COVID-19 may have affected people in lower socioeconomic groups worse than the affluent. The SARS-CoV-2 seroprevalence and the specificity of anti-SARS-CoV-2 antibody tests in South Africa is not known.

### Methods

We tested 405 volunteers representing all socioeconomic strata from the workforce of a popular shopping and tourist complex in central Cape Town with the Abbott SARS-CoV-2 IgG assay. We assessed the association between antibody positivity and COVID-19 symptom status, medical history, and sociodemographic variables. We tested 137 serum samples from healthy controls collected in Cape Town prior to the COVID-19 pandemic, to confirm the specificity of the assay in the local population.

### Results

Of the 405 volunteers tested one month after the first peak of the epidemic in Cape Town, 96(23.7%) were SARS-CoV-2 IgG positive. Of those who tested positive, 46(47.9%) reported no symptoms of COVID-19 in the previous 6 months. Seropositivity was significantly associated with living in informal housing, residing in a subdistrict with low income-per household, and having a low-earning occupation. The specificity of the assay was 98.54% (95%CI 94.82%-99.82%) in the pre-COVID controls.

**Funding:** The project was co-funded by the V&A Waterfront Pty, Ltd and the Stellenbosch University Immunology Research Group.

**Competing interests:** The authors have declared that no competing interests exist.

## Conclusions

There is a high background seroprevalence in Cape Town, particularly in people of lower socioeconomic status. Almost half of cases are asymptomatic, and therefore undiagnosed by local testing strategies. These results cannot be explained by low assay specificity.

## Introduction

The first case of Coronavirus disease-2019 (COVID-19) in South Africa was documented on March 5[th], 2020. Despite the early implementation of a stringent nationwide lockdown protocol, within 6 months South Africa (with a population of ¬60 million people) had registered over 620 000 cases of COVID-19, recording over 12 000 new cases per day at the peak of the first wave [1, 2]. The Western Cape Province of South Africa, a highly frequented tourist destination, suffered an earlier peak than the country's other provinces following several introductions of SARS-CoV-2 from Europe [3].

The City of Cape Town in the Western Cape Province is characterised by a startlingly high degree of inequality, with a Gini Index of 0.64 in 2014 [4]. Densely populated informal settlements, where residents live in shacks made of corrugated iron sheeting and are fully reliant on public transport, are found a few kilometres away from upscale leafy suburbs. During this epidemic, the inequalities between these areas have intensified. Many of the 'hotspots' of COVID-19 infections coincide with areas with lower per-capita income, and statistics show that the government-imposed containment measures were less effective there than in the more affluent suburbs [5].

Moreover, the number of reported cases does not reflect the true number of infections for several reasons. The national case definition relies on the identification of genetic material of SARS-CoV-2 on a respiratory specimen by reverse transcriptase-polymerase chain reaction (RT-PCR), a test with a sensitivity in the range of 80% [6]. Local testing strategies were hampered by limited capacity and have mostly excluded people without symptoms [7]. The proportion of asymptomatic cases is unknown, and estimates vary widely depending on the setting and the test format. Recently published seroprevalence studies confirm that a testing strategy which focuses purely on RT-PCR is likely to miss a large proportion of SARS-CoV-2 infections [8–10].

A recent sentinel surveillance study using residual blood samples from people attending routine antenatal and HIV care at public sector clinics in the Cape Town metropole, found SARS-CoV-2 seroprevalences ranging from 30.6% to 46.2% across all subdistricts [11]. This suggests that certain areas and population groups in Cape Town have experienced a very pervasive first wave. However, the specificity of SARS-CoV-2 antibody assays has not been assessed in this population, although results from testing of pre-COVID 19 plasma samples from Tanzania and Zambia suggest a high degree of false positives may be expected [12]. The seroprevalence of people from higher income areas, who rarely access public sector healthcare, is also not known. Here we report the results of a targeted workplace seroprevalence study in a popular shopping and tourist destination, which included people from all sociodemographic strata of Cape Town, as well as a control group from blood collected before November 2019.

## Methods

Between 17 August and 4 September 2020, we recruited a diverse group of volunteers from all levels of employment of the Victoria and Alfred Waterfront (V&A), which is South

Africa's most-visited shopping and tourist destination in Cape Town, South Africa. This timepoint was approximately one month after the peak of the first wave of COVID-19 cases in this region and 6 months after the first documented case in the country. At the time of testing, South Africa was in the process of deescalating stringent lockdown measures which had already been in force for 5 months. During the lockdown, most people did not attend work, either part time or full time, or worked fewer days or hours, unless designated as an 'essential service' worker.

The workforce from which our volunteer sample was drawn included approximately 1200 workers either directly employed by the V&A Waterfront, by one of the various museums and attractions on site, or by one of the companies acting as service providers. The volunteers were contacted and scheduled through their internal business communications structures. Volunteers were excluded if they had any symptoms of or known exposure to COVID-19 in the previous 10 days. Each participant attended a single visit in which they underwent informed consent counselling, provided written informed consent, and data on vitals and metrics, medical history, and COVID-19 exposure and symptoms were captured directly into an electronic database.

Blood collected from each participant was centrifuged, the serum divided into aliquots and frozen at -80˚C. Samples were analysed after a single freeze-thaw cycle with the Abbott SARS-CoV-2 IgG assay (Abbott Laboratories, South Africa, Pty Ltd) on the Architect i System (Abbott Laboratories, Illinois, USA), a chemiluminescent microparticle immunoassay (CMIA), for the qualitative detection of IgG antibodies to the SARS-CoV-2 nucleocapsid protein. An index of ≥1.4 was considered positive according to the manufacturer's specifications. This assay has a reported sensitivity of between 87.5% and 100%, and a specificity of between 99.63% and 100% [13–17]. Using this 99% specificity, a power of 80%, a significance level of 0.05 and a non-inferiority limit of 3%, we calculated that a sample size of 137 pre-COVID controls would be sufficient to ascertain non-inferiority of the Abbott SARS-CoV-2 IgG in the local population compared to the populations in which it has previously been tested and developed. We therefore tested 137 serum samples from asymptomatic controls from a low socio-economic status local community, collected before November 2019.

Analysis was conducted using R. Continuous variables were expressed as median and inter-quartile range and tested for normality with the Shapiro-Wilk Test. Continuous variables with Gaussian distribution were analysed with a Student's t-test, and non-Gaussian variables with a Wilcoxon test. Categorical variables were compared using a Chi-squared test with continuity correction. Fisher's exact test was used to confirm significance, which was set at $p < 0.05$. This study was approved by the Stellenbosch University Human Research Ethics Committee (reference N20/07/038-COVID-19).

## Results of the study sample

Out of 410 volunteers screened, five (four with symptoms compatible with possible COVID-19 on the day, and one with an episode of syncope prior to blood draw) were excluded from the study. The details of the participants are presented in Table 1, stratified by their SARS-CoV-2 serology result.

The sample population consisted mostly of healthy adults, with a median age of 38 years, and median Body Mass Index (BMI) of 28 (range 16.5–57.6). Hypertension was the most common comorbidity, with 57 (14.1%) of participants having a pre-existing condition, and a further 125 participants had persistently elevated blood pressure readings on the day (>140/90mmHg). There were 19 (4.7%) known diabetics, with 3 new diabetics diagnosed on site by point-of-care blood glucose and HbA1c testing. There were 28 (6.9%) participants who self-

**Table 1. SARS-CoV-2 seroprevalence, comorbidities, COVID-19 symptoms and exposure.**

| Demographic data | Total (n = 405) | Antibody Positive (n = 96) | Antibody Negative (n = 309) | p |
|---|---|---|---|---|
| Age (years) | 38 [18–69] | 36 [20–65] | 39 [18–69] | 0.108 |
| Female gender (n,%) | 217 (53.6) | 54 (56.2) | 163 (52.8) | 0.560 |
| Comorbidities (n,%) | | | | |
| None | 184 (45.4) | 52 (54.1) | 132 (42.7) | 0.231 |
| Hypertension | 57 (14.1) | 15 (15.6) | 42 (13.6) | 0.873 |
| BP persistently >140/90mmHg | 125 (30.9) | 36 (37.5) | 89 (22.0) | 0.302 |
| Diabetes | 22 (5.4) | 10 (10.4) | 12 (3.9) | 0.040[+] |
| HIV | 28 (6.9) | 9 (9.4) | 19 (6.2) | 0.375 |
| Asthma | 29 (7.2) | 3 (3.1) | 26 (8.4) | 0.078 |
| High Cholesterol | 21 (5.2) | 4 (4.2) | 17 (5.5) | 0.618 |
| Malignancy | 6 (1.5) | 1 (1.0) | 5 (1.6) | 1.000 |
| Heart disease | 7 (1.7) | 2 (2.1) | 5 (1.6) | 1.000 |
| Hypothyroidism | 7 (1.7) | 2 (2.1) | 5 | 1.000 |
| Reflux | 9 (2.2) | 1 (1.0) | 8 | 0.463 |
| Autoimmune disease | 4 (1) | 0 (0.0) | 4 (1.3) | 0.577 |
| Kidney disease | 3 (0.7) | 0 (0.0) | 3 (1.0) | 1.000 |
| Previous Tuberculosis | 30 (7.4) | 8 (8.3) | 12 (3.9) | 0.120 |
| Ever smoker | 152 (37.5) | 25 (26.0) | 127 (41.1) | 0.006[+] |
| **Symptoms of COVID-19 in the previous 6 months (n,%)** | | | | |
| Any symptoms | 180 (44.4) | 50 (52.1) | 130 (42.1) | NA |
| Specific symptoms | | | | |
| New cough | 86 (21.2) | 25 (26.0) | 61 (19.7) | 0.200 |
| Fever or chills | 67 (16.5) | 28 (29.2) | 39 (12.6) | <0.001 |
| Muscle aches | 59 (14.6) | 21 (21.9) | 38 (12.3) | 0.030[+] |
| New dyspnoea | 33 (8.1) | 11 (11.5) | 22 (7.1) | 0.200 |
| Sore throat | 76 (18.8) | 16 (16.7) | 60 (19.4) | 0.654 |
| Loss of smell | 31 (7.7) | 23 (24.0) | 8 (2.6) | <0.001 |
| Loss of taste | 32 (7.9) | 21 (21.9) | 11 (3.6) | <0.001 |
| Diarrhoea | 34 (8.4) | 7 (7.3) | 27 (8.7) | 0.833 |
| Nausea and/or vomiting | 11 (2.7) | 1 (1.0) | 10 (3.2) | 0.471 |
| **SARS-CoV-2 exposure (n,%)** | | | | |
| Contact with a COVID-19 case in the past 6 months | 70 (17.3) | 14 (14.6) | 56 (18.1) | 0.296 |
| Travel since December 2019 | | | | 0.068 |
| • Outside South Africa | 17 (4.2) | 0 (0.0) | 17 (5.5) | |
| • Within South Africa | 87 (21.5) | 17 (17.7) | 70 (22.7) | |

Participants' details stratified by their SARS-CoV-2 IgG antibody result, and factors associated with a positive antibody test. Age data are presented as median and range. BP, blood pressure; HIV, Human Immunodeficiency Virus; COVID-19, coronavirus disease 2019.

[+]These p values became non-significant after adjusting for the multiple testing effect via the Holm method.

declared as living with HIV, 24 of whom were on fixed-dose combination antiretroviral treatment. Other comorbidities are listed in Table 1.

Of the 405 participants included in the analysis, 96 (23.7%) tested positive for SARS-CoV-2 IgG antibodies. On multivariate analysis, having diabetes and being an 'ever smoker' were associated with testing SARS-CoV-2 antibody positive, but became non-significant after correction for the multiple testing effect. Overall, 180/405 (44.4%) reported having had symptoms of possible COVID-19 in the preceding 6 months. Of those who reported symptoms, 50/180

**Table 2. SARS-CoV-2 seroprevalence and sociodemographic indicators.**

| Sociodemographic indicators | Total | Antibody positive (n,%) | Antibody negative (n,%) | p |
|---|---|---|---|---|
| **Dwelling type**: | | | | 0.003[+] |
| Informal housing | 84 | 32 (38.1) | 52 (61.9) | |
| Formal housing | 321 | 64 (19.9) | 257 (80.1) | |
| **Employment type**: | | | | |
| Management | 54 | 3 (5.6) | 51 (94.4) | <0.001 |
| Administration and Support | 150 | 19 (12.7) | 131 (87.3) | <0.001 |
| Parking and Security | 50 | 16 (32.0) | 34 (68) | 0.156 |
| Housekeeping services | 136 | 57 (41.9) | 79 (58.1) | <0.001 |
| Other | 15 | 1 (6.7) | 14 (93.3) | 0.211 |
| 'Essential services' designation[*] | 68 | 20 (29.4) | 48 (70.6) | 0.273 |
| **District of residence (% of households with income <$10 per day)**: | | | | |
| Khayelitsha (49%) | 45 | 22 (49) | 23 (51) | <0.001 |
| Mitchells Plain (42.3%) | 37 | 12 (32.4) | 25 (67.6) | 0.226 |
| Klipfontein (38.8%) | 34 | 11 (32.4) | 23 (67.6) | 0.292 |
| Tygerberg (28.4%) | 41 | 11 (26.8) | 30 (73.2) | 0.700 |
| Southern (23.4%) | 106 | 22 (20.8) | 84 (79.2) | 0.426 |
| Western (24.7%) | 63 | 5 (7.9) | 58 (92.1) | <0.001 |
| Northern (21.1%) | 41 | 8 (19.5) | 33 (80.5) | 0.566 |
| Eastern (13.5%) | 24 | 3 (12.5) | 21 (87.5) | 0.222 |
| Cape Winelands, Overberg and West Coast | 5 | 1 (20.0) | 4 (80.0) | 1.00 |
| Unknown | 9 | 1 (11.1) | 8 (88.9) | 0.692 |

Sociodemographic indicators of participants stratified by their SARS-CoV-2 antibody result. District of Residence is listed in order of increasing income estimate, described by the percentage of households in that district with an income of less than $10 per day.

[*]Individuals who continued to attend work on site throughout all levels of lockdown.

[+]This p value became non-significant after adjusting for the multiple testing effect via the Holm method.

(27.8%) tested antibody positive. Of the symptoms reported, only fever, muscle aches, loss of taste and loss of smell were significantly associated with a positive antibody test on multivariate analysis. There was no association between close contact with a known COVID-19 case and a positive antibody result. Only 17 of the 405 participants travelled outside of South Africa after December 2019 and none tested positive for the SARS-CoV-2 antibody. There was no significant association between testing positive and travel within South Africa. Sixty-seven of all 405 participants reported having undergone RT-PCR testing, of whom 15 (22.4%) reportedly tested positive. Of these 15, ten tested positive for SARS-CoV-2 antibodies. In four of the five who reported testing RT-PCR positive but were SARS-CoV-2 antibody negative, the tests occurred more than 50 days apart (range 50–87 days); the fifth was tested with RT-PCR 21 days before antibody testing. Of those who tested antibody positive and had had RT-PCR testing, 12 reported a negative PCR test result.

Three interlinked sociodemographic variables were identified which correlated to a positive antibody test: the participant's district of residence, their type of dwelling or housing, and their occupation (Table 2).

Participants were stratified by health district of origin (in order to compare to reporting of locally registered cases). Districts were further defined by the percentage of households in each area with an income of less than $10 per day, based on the most recent census data [18]. The highest number of seropositive participants came from the Khayelitsha district (49% of households with income <$10 per day; SARS-CoV-2 seroprevalence 22/45, 49%, p<0.001).

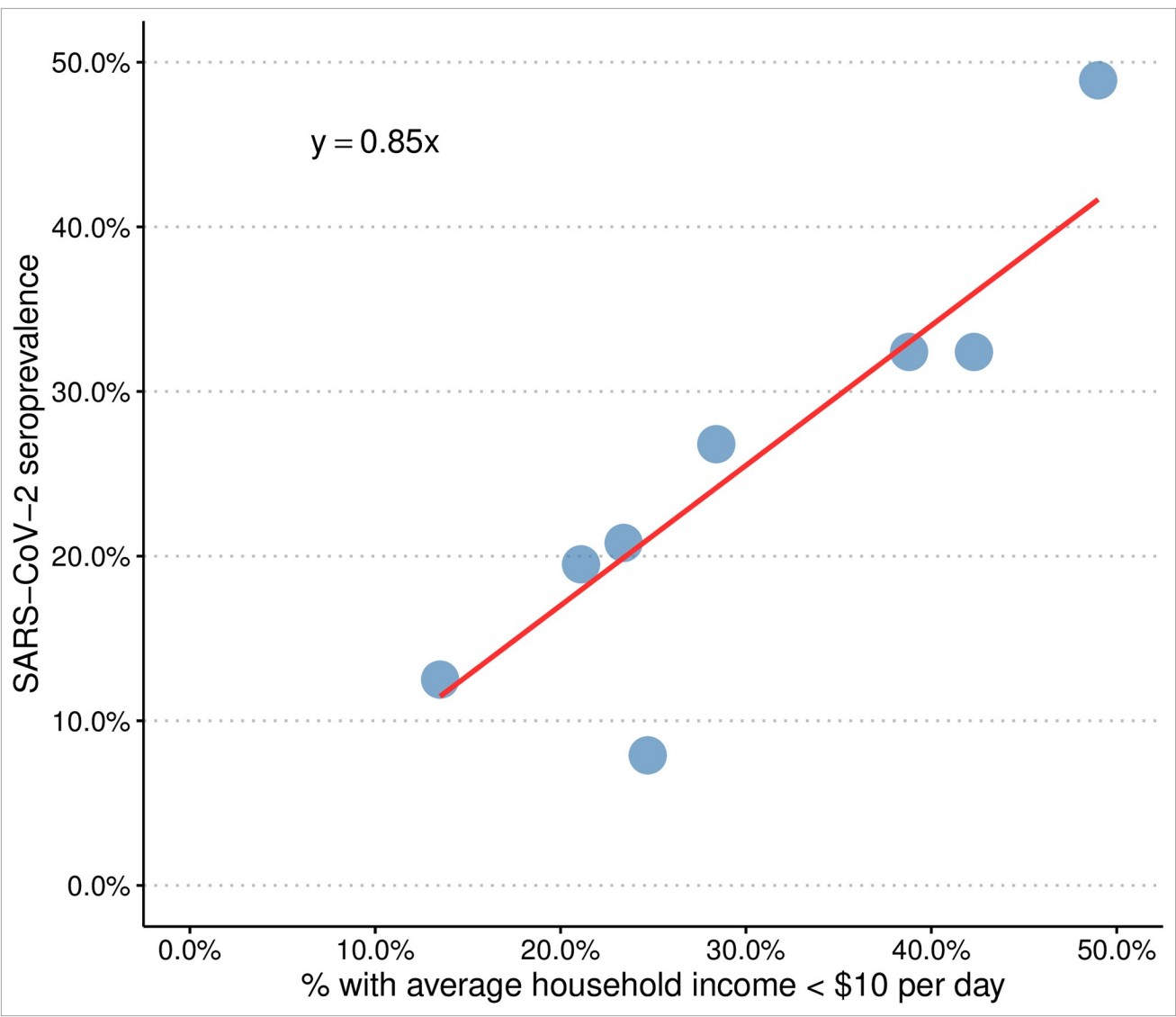

**Fig 1. The correlation between the income and the SARS-CoV-2 seroprevalence within the subdistrict of residence of the study population.** Income is denoted by the proportion of households with a daily income of <$10, with an increasing value implying a lower socioeconomic status.

Participants from the Southern district (23.4% of households with income <$10 per day) made up the largest group within the sample (26.8% of participants) and had a seroprevalence of 22/106, 20.8% (p = 0.426). Participants from the Western district (24.7% of households with income <$10 per day) were significantly more likely to test negative and had a seroprevalence of 5/63 (7.9%) (p<0.001). Fig 1 shows the relationship between the average income of the household in the district and the seroprevalence. Participants who lived in an informal dwelling were more likely to test positive, and those who live in a formal dwelling (house or flat) more likely to test negative (p = 0.001). When stratified by occupation, the largest proportion of the participants who tested positive arose from housekeeping services (cleaners), whereas the participants who worked in management were least likely to test SARS-CoV-2 antibody positive (p<0.001) (Fig 2).

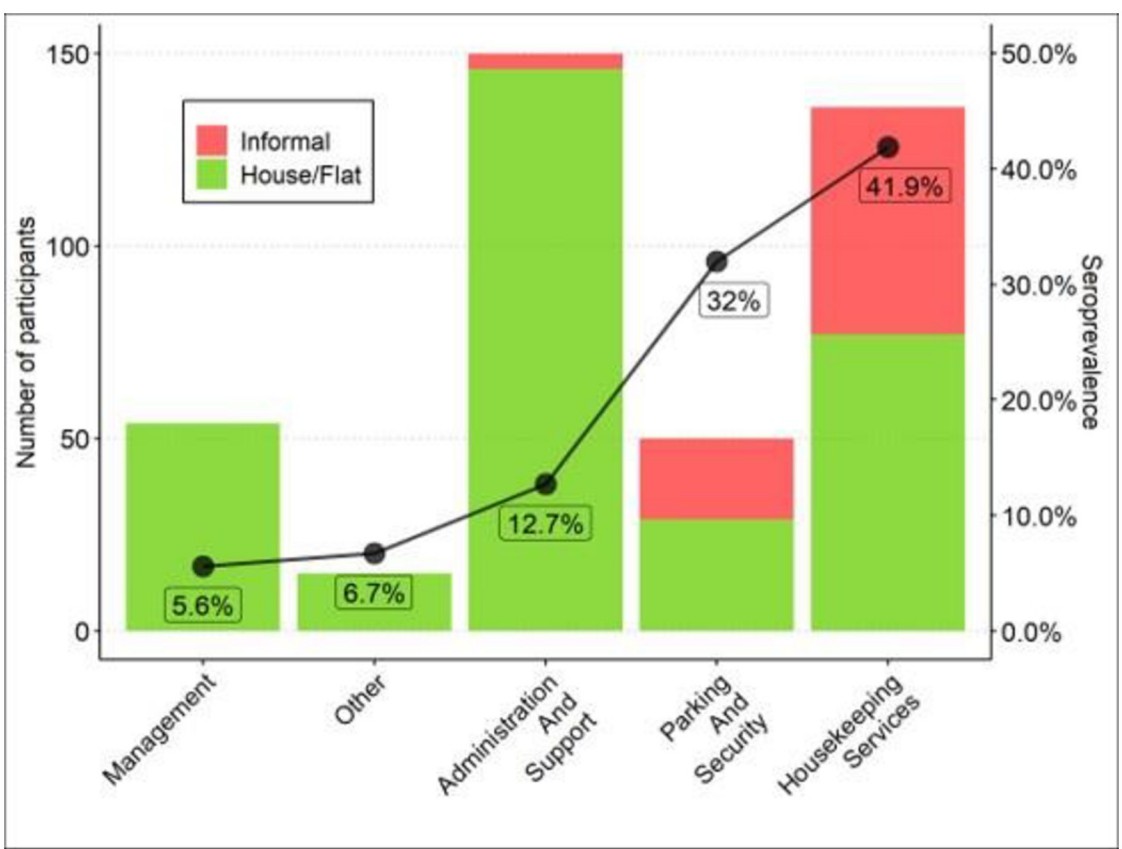

**Fig 2. The association between participants' occupations, type of dwelling and SARS-CoV-2 seroprevalence.** There is a higher seroprevalence in lower-earning occupation groups with a higher proportion of participants living in informal housing.

On sub-analysis of the participants who tested antibody positive, no significant association was found between reporting symptoms of COVID-19 in the previous 6 months or not and any demographic variables, smoking status, BMI, or medical conditions. On analysing medication use (including ACE-inhibitors, angiotensin receptor blockers, hydroxychloroquine, statins, oral and inhaled corticosteroids, antiretrovirals and multivitamin supplements), a significant association between statin use and symptomatic state was identified ($p = 0.018$), which became non-significant on correction for multiple testing effect ($p = 0.546$).

Sensitivity analysis was performed on the seven variables which were significantly associated with a positive serology result, using five scenarios with varying proportions (5.0%, 10%, 15%, 20% and 25.0%) of variables intentionally misclassified, with the harmonic mean p value method (HMP) used to identify a new p value for each variable. A single variable ('muscle aches') became non-significant with intentional misclassification.

## Results of the pre-COVID-19 controls

Of the 137 pre-COVID-19 serum samples analysed, two tested positive. Repeat testing of these two samples on the VIDAS SARS-CoV-2 IgG assay (bioMérieux, Midrand, South Africa) produced negative results for both. Assuming the absence of SARS-CoV-2 from the local population at the time of sample collection (before November 2019), this implies a specificity of 98.54% (95% CI 94.82%-99.82%, $p<0.05$) for the Abbott SARS-CoV-2 IgG Assay in this population when using the manufacturer's recommended cut-off of ≥1.4. Compared to the

published specificity of 99.6% for this assay, this test demonstrated non-inferiority in our local population with a margin of 2.75% at a power of 0.8 and significance level of 0.05.

## Discussion

This study of a targeted workplace with high exposure to international visitors during the early phases of the first wave of the epidemic demonstrates that people in lower socioeconomic groups of Cape Town have a higher seroprevalence of SARS-CoV-2 antibodies. Seroprevalence in participants' district of residence was strongly correlated with the percentage of people in the district with a low standardised measure of income. Moreover, participants in lower-earning occupations and those living in informal housing were more likely to test positive. Almost half of the participants from Khayelitsha—a partially-informal township in Cape Town afflicted by overcrowding and poverty—tested positive for SARS-CoV-2 antibodies.

Interestingly, there was no correlation between antibody positivity and so called 'essential services' designation. This implies that, despite working in an environment with high exposure to international tourists before lockdown, and the risks of continued use of public transport during lockdown, those infected were probably infected at home rather than the workplace. However, it is likely that this is true for only the categories of essential services included in this study, as healthcare workers are likely to have a higher seroprevalence [19, 20]. Rather, our data highlights the fact that some people are less able to comply with non-pharmacological interventions to prevent transmission than others because of their living circumstances. Similar inequalities have been documented in high-income countries: in New York City, the number of COVID-19 cases detected by RT-PCR was significantly associated with multiple socioeconomic indicators, including population density, median household income, and dependent children [21]. Another study compared the number of proven cases and deaths due to COVID-19 in each county of the USA to a poverty index. They found that early in the pandemic, the counties with a higher poverty index had a higher number of cases, and throughout the pandemic, these areas had a higher number of deaths due to COVID-19 [22]. In Leicester, UK, a larger household size and belonging to an ethnic minority were both associated with a higher likelihood of testing SARS-CoV-2 RT-PCR positive [23]. Jay and colleagues used smartphone tracking data in the USA to demonstrate that the 'stay at home' orders were only associated with smaller increases in staying home in low-income neighbourhoods compared to high-income neighbourhoods [24]. Our participants from low-income districts face multiple physical barriers to the social distancing and stay-at-home orders, and these may explain the higher seroprevalence in these areas.

The overall seroprevalence in the current study of 23.7% after the first wave is higher than in countries with a higher average income, lower inequality and an ability to shelter their population, such as Germany and Ireland [25]. However, our result is similar to a sample from a pilot study in Niger State, Nigeria (Gini index 0.35) which reported a seroprevalence of 25.4% independent of rural or urban residence [26], and to the studies which reported higher seroprevalences in more densely populated areas within countries. For example, the overall seroprevalence of a convenience sample from New York State in the USA was 12.5%, but in New York City itself, where population density and inequality between neighbourhoods is higher, the estimated cumulative incidence of COVID-19 cases was 22.7% [27]. A Spanish seroprevalence survey found an overall seroprevalence of only 4.7%–5% in May of 2020, which increased to 14.4% in the more densely populated central provinces [9]. A recent survey of 3098 Kenyan blood donors found an overall seroprevalence of 5.6% (population-weighted seroprevalence 4.3%), which was highest in those living in the three largest urban counties of Mombasa

(8.0%), Nairobi (7.3%) and Kisumu (5.5%) [28]. Similar to our study, their results suggested that infection in Kenya was more widespread than the current PCR-based testing strategies would suggest.

In our study, almost half of all positive participants were completely asymptomatic of COVID-19 in the previous 6 months, suggesting that socioeconomic status of participants was not associated with higher severity of disease. Even though the prevalence of uncontrolled or undiagnosed comorbidities which may predispose to severe COVID-19 is expected to be higher in areas of low socioeconomic status, this group may not be part of the active workforce and so were not included in our sample. The seropositive-asymptomatic proportion in our sample is similar to that in other studies. The Spanish study reported that between 28.5% and 32.7% of all who tested positive had not had symptomatic COVID-19 [9]. In an Icelandic study, 41 (3.3%) of 1244 people in quarantine for SARS-CoV-2 exposure tested antibody positive, of which 24 (58.5%) reported no symptoms [10]. The high proportion of asymptomatic infections contributes to the case detection and reporting gap, along with patients with mild symptoms often not presenting to healthcare at all, and the effect of the highly targeted testing strategies employed in South Africa due to limited testing capacity.

The presence of SARS-CoV-2 antibodies in serum has not yet been definitively correlated with long term immune memory. However, evidence of a high baseline seroprevalence (at least in some regions) is encouraging as it could mean that communities less shielded from infection pressure for socioeconomic reasons may at least be less affected an overwhelming 'second wave', barring any significant mutations to the virus which impact the efficacy of neutralising antibodies.

Our study found an unexpected association between reporting symptoms in keeping with COVID-19 in the previous 6 months and the use of statins, which have previously been associated with a reduced severity of COVID-19 [29]. It's possible that our sample method selected out those who suffered severe COVID-19, and that statins should rather be considered markers of comorbidity (and therefore higher risk for symptomatic COVID-19) in this population.

In our local pre-COVID-19 control group, the specificity of the assay used in our study was non-inferior to the reported specificity, supporting the reliability of our findings. An important limitation of this study is the moderate sample size which diminished the statistical significance of many of the findings. Furthermore, the design of the study was such that we could only sample selected participants from each sociodemographic and geographic group, who were not necessarily representative of the overall subpopulation they arose from. A further limitation of this sampling method is that it resulted in unequal numbers from each district and occupational group, which complicated comparisons between groups. In addition, our sample excluded those who are not part of the labour force, children, and the elderly and so can only be considered applicable to the economically active portion of the population. Our offer of serologic testing for SARS-CoV-2 before it was freely available may have introduced bias by selecting people who had a strong suspicion of having been infected. Nonetheless, we recruited a high proportion (405, 34%) of the potential pool of 1200 volunteers and our sample included participants from a broad range of sociodemographic backgrounds, less than half of whom ever experienced symptoms of possible COVID-19 and very few of whom had contact with a known case. In hindsight, it was felt that the participant's use of public versus private transport and the number of people in their household would have been useful sociodemographic indicators to analyse, but this data was not collected. Lastly, reported symptoms were fully reliant on participants' recollection, and we were unable to confirm the RT-PCR results of those participants who reported having the test, which precluded further meaningful analysis of this data.

## Conclusion and recommendations

In this study, we have shown that the Abbott SARS-CoV-2 IgG assay is highly specific in our population. We found a higher seroprevalence than most other published reports, but a similar proportion of asymptomatic infections. We have also demonstrated that the first wave of the local epidemic affected people of lower socioeconomic status worse than the affluent. Our recommendation to policy makers in South Africa and Sub-Saharan Africa in their planning for future waves is to manage their expectations of their population, and to avoid making inferences about one subpopulation from the seroprevalence data of another. In Cape Town, strict lockdown regulations were implemented, including restrictions on businesses, educational institutions, the closure of borders to restrict traveling and a ban on all social and religious gatherings. Public health measures like the wearing of masks and an emphasis on regular sanitization of hands and surfaces were widely publicised and legally enforced. Despite this, these measures seem not to have benefitted socio-economically poorer population groups. Poor housing conditions, including crowding, and the lack of living spaces that would allow social distancing for several weeks or months, possibly contributed to a rapid spread of COVID-19 infection amongst low socio-economic strata in the population. Policy makers need to take into account that it is not reasonable to expect people whose dwellings are barely adequate as sleeping quarters and that lack any recreational space, to practice social distancing to the same degree as the better off sectors of society. Furthermore, cultural factors which drive social interactions in these areas must also be understood. This study was not designed to investigate alternative COVID-19 control measures, but the social and cultural aspects of socio-economically deprived areas will have to be considered in future epidemics when control measures are planned. It would appear, that for the COVID-19 pandemic at least, the only likely way to protect the socioeconomically vulnerable is to ensure an early and effective vaccination strategy.

## Acknowledgments

We extend our thanks to the V&A Waterfront for proposing the project, and to Mr Andre Theys, his administrative team, and the V&A community advisory board for the warm welcome and continuous support throughout its execution.

## Author Contributions

**Conceptualization:** Jane Alexandra Shaw, Tracy Cummins, Novel N. Chegou, Nelita Du Plessis, Léanie Kleynhans, Vinzeigh Leukes, Andre G. Loxton, Helmuth Reuter, Donald Simon, Wolfgang Preiser, Stephanus T. Malherbe, Gerhard Walzl.

**Data curation:** Jane Alexandra Shaw, Maynard Meiring, Kim Stanley.

**Formal analysis:** Jane Alexandra Shaw, Maynard Meiring, Gerard Tromp.

**Funding acquisition:** Helmuth Reuter, Gerhard Walzl.

**Investigation:** Jane Alexandra Shaw, Tracy Cummins, Marika Flinn, Andriette Hiemstra, Candice MacDonald, Nosipho Mtala, Stephanus T. Malherbe.

**Methodology:** Jane Alexandra Shaw, Tracy Cummins, Conita Claassen, Vinzeigh Leukes, Gerard Tromp, Stephanus T. Malherbe.

**Project administration:** Jane Alexandra Shaw, Tracy Cummins, Conita Claassen, Andriette Hiemstra, Vinzeigh Leukes, Candice MacDonald, Nosipho Mtala, Donald Simon, Stephanus T. Malherbe.

**Resources:** Conita Claassen, Marika Flinn, Andriette Hiemstra, Candice MacDonald, Nosipho Mtala, Donald Simon.

**Software:** Kim Stanley.

**Supervision:** Jane Alexandra Shaw, Stephanus T. Malherbe, Gerhard Walzl.

**Writing – original draft:** Jane Alexandra Shaw.

**Writing – review & editing:** Tracy Cummins, Novel N. Chegou, Nelita Du Plessis, Léanie Kleynhans, Andre G. Loxton, Helmuth Reuter, Donald Simon, Wolfgang Preiser, Stephanus T. Malherbe, Gerhard Walzl.

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
