## [Decision Letter · Decision Letter 0]

3 Feb 2021

PONE-D-21-00967

Higher SARS-CoV-2 seroprevalence in workers with lower socioeconomic status in Cape Town, South Africa.

PLOS ONE

Dear Dr. Shaw,

Thank you for submitting your manuscript to PLOS ONE. After careful consideration, we feel that it has merit but does not fully meet PLOS ONE’s publication criteria as it currently stands. Therefore, we invite you to submit a revised version of the manuscript that addresses the points raised during the review process.

We look forward to receiving your revised manuscript.

Kind regards,

Godfrey Musuka

Academic Editor

PLOS ONE

Additional Editor Comments:

Many thanks for allowing me to review this key paper which will assist the response in southern Africa in many ways. Please address the comments from the other reviewers. and also address the following comments from me:

1. Please beef up substantially the description on the weakness of the study

2. Please provide detailed recommendations for policy makers in RSA and southern Africa as a whole.

2. Please provide additional details regarding participant consent. In the ethics statement in the Methods and online submission information, please ensure that you have specified:

 - whether consent was obtained

 - whether consent was informed

 - what type of consent you obtained (for instance, written or verbal, and if verbal, how it was documented and witnessed).

"We extend our thanks to the V&A Waterfront for proposing and funding the project, and to

295 Mr Andre Theys, his administrative team, and the V&A community advisory board for the

296 warm welcome and continuous support throughout its execution."

"The authors received no specific funding for this work."

Reviewers' comments:

Reviewer's Responses to Questions

**Comments to the Author**

1. Is the manuscript technically sound, and do the data support the conclusions?

Reviewer #1: Yes

Reviewer #2: Yes

2. Has the statistical analysis been performed appropriately and rigorously? 

Reviewer #1: Yes

Reviewer #2: Yes

3. Have the authors made all data underlying the findings in their manuscript fully available?

Reviewer #1: Yes

Reviewer #2: Yes

4. Is the manuscript presented in an intelligible fashion and written in standard English?

Reviewer #1: Yes

Reviewer #2: Yes

5. Review Comments to the Author

Reviewer #1: This is an interesting article and informs policy formulation in combating Covid-19. The paper looks at two aspects, firstly the SARS-CoV-2 seroprevalence in workers with lower socioeconomic status in Cape Town and then checking the specificity of the Abbott SARS-CoV-2 IgG Assay. These are very important issues as the whole world suffers from the Covid-19 pandemic.

In the first part of the study, the authors tested 405 volunteers representing all socioeconomic strata from the workforce of a popular shopping and tourist complex in central Cape Town with the Abbott 29 SARS-CoV-2 IgG assay. They assessed the association between antibody positivity and COVID- 19 symptom status, medical history, and sociodemographic variables.

The study found that Seropositivity was significantly associated with living in informal housing, residing in a subdistrict with low income-per household, and having a low-earning occupation. The main strength of this paper is that it addresses an interesting and timely question of whether there is an association between antibody positivity and COVID- 19 symptom status, medical history, and sociodemographic variables. It is a pertinent issue in Africa as most families have low-income jobs and live in high densely populated towns. Therefore, it was a noble idea to find out if there was a link between lower socioeconomic and SARS-CoV-2 seroprevalence.

In the second part of the study, the specificity of the Abbott SARS-CoV-2 IgG Assay. The study also noted a high specificity of the assay with regard the South African population. The sample size for the verification exercise validates the results of the assay specificity. However, it would have been good to compare the serum sample results with SARS-CoV-2 RT-PCR as it is currently the gold standard instead of assuming the absence of SARS-CoV-2 from the local population at the time of sample collection. It was also ideal to check Sensitivity of the test kit to complete the process of kit verification. This would enhance the outcome of the study as the accuracy of the test kit in the given population would have been determined.

Overall, the paper is good and concise and brings valuable information.

Reviewer #2: In general, the paper is logically presented and is easy to follow. A few spelling and grammar mistakes are present as outlined in my comments below.

The title and abstract cover the main aspects of the work. The results uncover the high seroprevalence that are not detected by the current strategies, this is novel. This provides useful insights for strategies review. The methods are clear and replicable.

All the results presented match the methods described and statistical analysis were appropriate to the research question and study design. Data presentation was clear and easy to follow. Control selection was appropriate for the study design.

There is correlation of results and conclusion. I am satisfied with the validity of the manuscript.

A. Major Comment

1. No attempt has been made to address potential bias through analytic methods, eg., sensitivity analysis.

B. Minor Comments.

1. Line 99: It is important to write how serum is obtained since you can not directly collect serum from a patient.

2. Line 148: It is not clear if these 17 participants are out of the 180 or the whole sample of 405.

3. Line 151: It is not clear if these 67 are out of the total sample or 180, make it clear.

4. L-211-212-213-214. From your study there has been no significant association between designated essential workers and sero-positivity. Is it possible that in the discussion of this finding you add and highlight some studies of sero-prevalence done among essential workers such as Healthcare workers as they have been found out to be associated with risk of infection from SARS-CoV 2?

5. This paper highlights the issues of socio-economic and demographic status in relation to SARS CoV 2 sero-positivity and as such did you by any means collect data on the races of the participants as it would be interesting to know and link to prior studies done of who is marginalised and of low socio-economic status in Western Cape and their perceived associated risk to SARS-CoV 2 infection.

6. The discussion section gave comparative analysis to UK and USA and Nigeria; please check if there are any other sero-prevalence studies in Africa, or Sub Sahara Africa that corroborates to your findings

C. PLOS ONE does not copyedit accepted manuscripts, so the language in submitted articles must be clear, correct, and unambiguous. Authors may consider the below small changes as applicable.

1. Line 23: South Africa has a “higher” degree not South Africa has a “high” degree since this is comparison.

2. Line 40: Should instead be “Almost half of cases are asymptomatic” not “Half of cases are asymptomatic” because it is not exactly 50% according to your findings.

3. Line 82: Put comma (,) between “2020” and “we”.

4. Line 88: should read “During the stringent lockdown” add “the”.

5. Line 88: Add comma between “lockdown” and “most”.

6. Line 88-89: Add comma between “work” and “either.

7. Line 98: should read “directly into” not “directly “on to”.

8. Line 101: Add these commas “Abbott Laboratories, South Africa, Pty Ltd”.

9. Line 103: Should be in past tense and read “An index of >1.4 was considered” not “is considered”.

10. Line 105: Should read “of between 87.5% and 100%”,

11. Line 105: Should read “of between 99.63% and 100%

12. Line 105: should read “Using a 99% specificity” not “Using this 99% specificity”.

13. Line 107: consider the word “ascertain” not “assert”.

14. Line 121-122: Should read: Out of 410 volunteers screened, 5 (4 with symptoms compatible with possible COVID-19 on the day, and 1 with an episode of syncope prior to blood draw) were excluded from the study

15. Line 134: rather “pre-existing condition” not “pre-existing diagnosis”.

16. Line 143: remove “any” from “having had any”.

17. Line 188: put a comma between the closing bracket and a

18. Line 204: Remove comma between “epidemic” and “demonstrates”.

19. Line 218: Put a comma between “City” and “the”

20. Line 222: Put comma between “pandemic” and “the counties”.

21. Line 223: Put comma between “pandemic” and “these”.

22. Line 223: Put “a” between “had” and “higher”.

23. Line 224: Put “a” between “UK,” and “larger”.

24. Line 227: Should be “smaller increases” not “small increases” since this is comparison.

25. Line 231: Remove comma between “wave” and “is higher”.

26. Line 232: add “a” between “with” and “high”.

27. Line 232” use “higher” and “lower” since this is a comparison.

28. Line 242: Add comma between “study” and “almost”.

29. Line 250: Add comma between “study” and “41”.

30. Line 251: Rather make it read “of which 24” instead of “of whom 24”..

31. Line 265: Should be “markers” instead of “marker” since statins is plural too.

32. Line 267: Put comma between “group” and “the”.

33. Line 269: Remove comma between “method” and “which”

34. Line 284: Put comma between “study” and “we”

6. PLOS authors have the option to publish the peer review history of their article (what does this mean?). If published, this will include your full peer review and any attached files.

Reviewer #1: No

Reviewer #2: **Yes: **Dr. Tafadzwa Dzinamarira

---

## [Author Response · Author response to Decision Letter 0]

12 Feb 2021

We have attached a 'response to reviewers' letter which responds to all comments from the reviewers and editor. We also attach revised manuscripts with and without tracked changes. 

The figures were uploaded to, and reviewed by PACE before the first submission, and so this process has not been repeated. 

Thanks for the opportunity.

---

## [Editor Report · Decision Letter 1]

15 Feb 2021

Higher SARS-CoV-2 seroprevalence in workers with lower socioeconomic status in Cape Town, South Africa.

PONE-D-21-00967R1

Dear Dr. Shaw,

We’re pleased to inform you that your manuscript has been judged scientifically suitable for publication and will be formally accepted for publication once it meets all outstanding technical requirements.

Kind regards,

Godfrey Musuka

Academic Editor

PLOS ONE
---

## [Editor Report · Acceptance letter]

18 Feb 2021

PONE-D-21-00967R1 

Higher SARS-CoV-2 seroprevalence in workers with lower socioeconomic status in Cape Town, South Africa. 

Dear Dr. Shaw:

I'm pleased to inform you that your manuscript has been deemed suitable for publication in PLOS ONE. Congratulations! Your manuscript is now with our production department. 

Kind regards, 

on behalf of

Dr. Godfrey Musuka 

Academic Editor

PLOS ONE